# Dental Care for Asylum-Seekers in Germany: A Retrospective Hospital-Based Study

**DOI:** 10.3390/ijerph17082672

**Published:** 2020-04-13

**Authors:** Anna Freiberg, Andreas Wienke, Lena Bauer, Andreas Niedermaier, Amand Führer

**Affiliations:** Interdisciplinary Center for Health Sciences, Institute for Medical Epidemiology, Biometrics and Informatics (IMEBI), Martin-Luther-University Halle-Wittenberg, Magdeburger Straße 8, 06112 Halle (Saale), Germany; andreas.wienke@uk-halle.de (A.W.); lena.bauer@gmx.com (L.B.); andreas@niedermaiers.com (A.N.); amand-gabriel.fuehrer@uk-halle.de (A.F.)

**Keywords:** secondary data analysis, asylum seekers, dental health care utilization, oral health

## Abstract

*Background:* The growing immigration to Germany led to more patients whose medical needs are divergent from those of the domestic population. In the field of dental health care there is a debate about how well the German health system is able to meet the resulting challenges. Data on asylum-seekers’ dental health is scarce. This work is intended to reduce this data gap. *Methods:* We conducted this retrospective observational study in Halle (Saale), Germany. We included all persons who were registered with the social welfare office (SWO) in 2015 and received dental treatments. From the medical records, we derived information such as complaints, diagnoses, and treatments. *Results:* Out of 4107 asylum-seekers, the SWO received a bill for 568 people. On average, there were 1.44 treatment cases (95%-CI: 1.34–1.55) and 2.53 contacts with the dentist per patient (95%-CI: 2.33–2.74). Among those, the majority went to the dentist because of localized (43.2%, 95%-CI: 38.7–47.7) and non-localized pain (32.0%, 95%-CI: 27.8–36.2). The most widespread diagnosis was caries (*n* = 469, 98.7%, 95%-CI: 97.7–99.7). *Conclusion:* The utilization of dental care is lower among asylum-seekers than among regularly insured patients. We assume that the low prevalence rates in our data indicate existing access barriers to the German health care system.

## 1. Introduction

### 1.1. Background

During the last years, a considerable increase in immigration occurred in Germany. Consequently, the growing immigration has led to more patients whose native language is not German and whose medical needs diverge from those of the domestic population [1]. This results from differing legal and socio-cultural determinants of health [2]. While engaging for their specific medical needs, new challenges for the healthcare system to integrate asylum-seekers arise. The last years saw a substantial increase in research connected to migration and health, but there is still a huge gap in the scientific knowledge about dental care.

### 1.2. State of the Science

This lack of scientific evidence is problematic due to the massive undersupply of dental health care, which has been reported by civil society actors [3]. In a statement on the care of asylum-seekers, the Bundeszahnärztekammer (BZÄK), therefore, points out that in the event of ethical conflicts, dentists must follow the ethical standards of the model professional code [4]. The debate on this complex of problems is made more difficult by the fact that the data situation on the care of asylum-seekers in Germany is generally patchy, especially in the dental field. Data on the dental health of asylum-seekers are available from Algeria, Canada, Australia, and the USA [5,6,7,8]. They show a high prevalence of caries, periodontitis, and gingivitis by asylum-seekers. An exploratory literature review in PubMed does not include many comparable studies from Germany. The exceptions are an unpublished dissertation on caries by children of asylum-seekers in Würzburg [9], a clinical trial on the oral health of asylum-seekers in Greifswald [10], and three oral health studies of refugees published in 2017 and 2018 [11,12,13]. All studies agree with the international literature in that caries are more frequent among asylum-seekers than in the corresponding age groups of patients of the receiving countries’ populations.

Data on the general use of the health care system by asylum-seekers are available from a number of European countries such as Germany, the Netherlands, and Ireland, and from Australia, the USA, and Canada [14,15,16,17]. Different legal frameworks seem to play a major role here, which is why the data cannot be easily generalized across countries.

### 1.3. Legal Context

In Germany, access of asylum-seekers to the health system is regulated by the Asylum-Seekers’ Benefits Act (AsylbLG). Therein, the range of services for the duration of the asylum procedure (or the first 15 months of stay in Germany) is limited in comparison to patients in the statutory health insurance (GKV), though the degree of limitations is a matter of dispute among lawyers [18]. While § 4 AsylbLG stipulates the coverage of costs for the treatment of (among others) “acute illnesses and pain” and explicitly excludes costs for prosthetic dentistry, § 6 AsylbLG allows the coverage of costs for “other services” if they are “necessary to maintain and safeguard health” [19,20].

In Saxony-Anhalt, where this study took place, asylum-seekers are not issued an electronic health insurance card (EHIC), as regularly insured patients are. They need a so-called medical treatment voucher (MTV) before they can use medical services. These are issued by the “administrative staff of the social welfare office” and assure physicians that treatment costs will be covered by the social welfare office [21]. Medical treatment vouchers (MTVs) for dental care are valid only once and for a short time and must be presented to the doctor. Follow-up consultations require a new MTV. The prescription of medication, as well as hospitalizations, need a written consent of the social welfare office in advance [22]. These administrative procedures build up barriers in access to dental health, since patients, as well as dentists and their practice teams often do not know how the care according to the Asylum-Seekers’ Benefits Act should be implemented [23]. In addition, the approach of social welfare offices in cost absorption differs from district to district and between federal states [24]. This creates an inconsistent situation in dental care for asylum-seekers, from which dentists and patients suffer. For the dentists it is not even always guaranteed that their services will be financially rewarded. This creates a complex and often unclear situation. Data on asylum-seekers’ dental care utilization can help to disentangle this complex situation.

### 1.4. Aims

The aim of this study is to provide an in-depth picture of the dental services utilized by asylum-seekers. Hereby, we aim to quantify the utilization of dental health services, the complaints leading to visits at the dental clinic, the diagnoses made by the dentist, and the treatments resulting from it.

## 2. Materials and Methods

This was a retrospective observational study. Before the commencement of data collection, ethics approval was sought from and granted by the Ethics Committee of the Medical Faculty of Martin-Luther-University Halle-Wittenberg (reference number: 2018-84).

### 2.1. Inclusion and Exclusion Criteria

This study included all persons who were registered with the social welfare office in Halle (Saale), Germany, as asylum-seekers at any time in 2015 for at least 1 day and received any dental care while registered as asylum-seekers. This information was generated based on billing data. The files of asylum-seekers fulfilling these inclusion criteria were retrieved pseudonymous based on their file number. All persons whose files could not be found were excluded.

### 2.2. Data Source

The reviewed period was from 1 January 2015–31 December 2015 at the Dental Department at Martin-Luther-University Halle-Wittenberg, Halle, Germany. The data were obtained from handwritten medical reports, which were available in paper form.

### 2.3. Data Collection

The data collection took place from April until August 2019. The medical reports were stored in the outpatient department and the central medical record archive. After anonymization, the data were entered into a SQL-database.

### 2.4. Variables

Date of birth and date of treatment was excerpted from the front page of the medical reports. The date of treatment was differentiated by ambulatory treatment at the Dental Department and treatments at the Emergency Department of a hospital. For each ambulatory dentist contact, information on the following questions was collected: The reasons for medical attendance, diagnoses, classified procedure codes (BEMA), prescribed pharmaceuticals, and if the patients kept the appointment or why it failed. The drugs were recorded according to the standard Anatomical Therapeutic Chemical (ATC) classification. The BEMA fee classification describes the standard of evaluation of dental services and forms the base for accounting within the statutory health insurance in Germany. Some medical reports contained additional information in terms of requests for cost reimbursement by the social welfare office. These included details about which treatment was requested and if the caseworker decided to approve or refuse the claim. In case of treatment at the Emergency Department, medical reports comprised a file note. The letter of referral informed about diagnoses, procedure codes, prescribed pharmaceuticals, and instructions for follow-up treatment.

### 2.5. Statistical Analysis

We reported absolute and relative frequencies and their 95% confidence intervals. Hereby different denominators were used: First, for all of the analyzed variables, we reported the frequencies in relation to all patients, i.e., all asylum-seekers who were eligible for inclusion in the study and whose file could be found. These frequencies displayed the prevalence of the variable in question for those who successfully gained access to the health care system. Since asylum-seekers face many barriers to access, it must be assumed that there are asylum-seekers who needed dental treatment but were not seen by a dentist. Therefore, the second type of frequency used all asylum-seekers as a denominator to estimate the minimal prevalence in the whole population of asylum-seekers.

We differentiated in the frequency of treatments between treatment cases and dentist contacts. One treatment case included all treatments of a patient due to the same medical problem. Therefore, one treatment case might comprise more than one dentist contact.

Since this was an exploratory approach, in line with current guidelines, no statistical tests were performed [25]. Analyses were performed in SAS (Cary, North 219, CA, USA).

## 3. Results

### 3.1. Demography

Out of the 4107 asylum-seekers potentially eligible for dental care in 2015, the social welfare office received a bill concerning dental care for 568 people. Of those, 93 files could not be found, thus overall, the files of 475 people were retrieved that formed the study sample. Thereof, some files were fragmentary (*n* = 25) or contained information from a year other than 2015 (*n* = 26). In sum, we analyzed 83.6% of the files of all asylum-seekers who received dental care in 2015, which equals to 11.6% of all asylum-seekers registered in 2015. Patients came from 29 different countries, most frequently from Syria (39.4%), Afghanistan (9.7%), Iran (8.0%), and Somalia (6.1%). More than 50% of the included asylum-seekers were between 20 and 34 years of age. Table 1 presents more demographic characteristics of the asylum-seekers.

### 3.2. Dental Health Care Utilization

In the year 2015, there were 1203 arranged appointments with a dentist as part of the 424 treatment cases. On average, there were 1.44 treatment cases (95%-CI: 1.34–1.55) and 2.53 contacts (95%-CI: 2.33–2.74) with the dentist per patient. The maximum treatment cases varied from 0–8. About 10.7% of health records contained information from a year other than 2015 or were fragmentary because, for example, patients left before they were treated (*n* = 51). With 59.4%, the largest number of patients came because of one dental health-related treatment case (*n* = 282). Less than 6.1% of the patients required more than 3 treatment cases. More details are given in Figure 1. Seventeen patients received at least once a dental treatment at the Emergency Department (3.6%).

In the year 2015, appointments failed in 217 cases. Reasons were unexcused absence (*n* = 53, 24.4%) and postponed appointments due to communication problems and the need to summon a translator (*n* = 55, 25.4%). In 26 cases (12.0%), patients who came without a medical treatment voucher were sent home. In addition, some patients left before they were treated (*n* = 14, 6.5%), were late (*n* = 10, 4.6%), or stopped the treatment for reasons of fear (*n* = 7, 3.2%). Other reasons were that the social welfare office had to be consulted first to clarify questions about the insurance status or about treatments to be approved; treatment cancellations, and others (6.5%, 4.6%, and 12.9%, respectively).

### 3.3. Clinical Presentations

#### 3.3.1. Discomfort

Localized and non-localized pain were the most common complaints leading to a consultation with the dentist (43.2% and 32.0%, respectively). Many treatment reasons were classified in the patient file as unspecified or unknown (30.7%). A large number (22.3%) of treatment appointments took place at the patient’s own request. Comparably few people came because of denture discomfort (4.2%) or with indolent disorders (10.7%). Other complaints were dental problems like insufficient fillings, eating difficulties, gingival bleeding, and temperature-sensitivity (7.8%, 4.8%, 4.0%, and 3.4%, respectively). More details are given in Figure 2.

#### 3.3.2. Diagnoses

The most widespread diagnosis was caries with 469 cases (98.7% of all patients). These were classified it into 8 sub-categories, depending on the recurrence and severity code. The most common form of the occurring caries was deep caries (*n* = 107, 22.5%). About 21% of all carious disease were classified as primary caries or as carious lesions (*n* = 100, *n* = 101). In relation to all patients, 96 cases of deep complicated caries (20.2%) were counted. In 275 cases (57.9%), patients showed poor teeth conditions. Furthermore, there were 139 cases (29.3%) of teeth deemed not worthy of preserving. 115 treatments (24.2%) were due to periodontitis. The two forms were differentiated, whereof apical periodontitis was most frequent (*n* = 80, 16.8% of all patients). In sum, 108 filling defects (22.7%) and 102 root canal prepared teeth (21.5%) were counted. In addition, 67 cases of pulpitis (14.1%) were registered, from which 64 were irreversible (13.5%). Fifty cases of accretion (10.5%) were recorded, composed of one half each plaque and calculus. In 37 medical examinations (7.8%), the doctors observed gingivitis. There were 32 treatments caused by inflammation (6.7%) like gumboils, fistulas, deep periodontal pockets, or cysts. The frequencies of different diagnoses are given in Table 2.

#### 3.3.3. Treatments

435 asylum-seekers had contact with the dental clinic and received at least one dental consultation or examination (10.6% of all asylum-seekers). During their therapy, at least 274 patients were administered local anesthetics (6.7%), and 169 patients were treated with root canal therapy (4.1%). In a minimum of 111 cases, surgical interventions were performed (2.7%), and 67 asylum-seekers received minimal interventions (1.6%).

In some fields, the frequencies of treatments differed between the genders. Of all men, only 55 got a prophylactic treatment (1.8%), whereas 35 of all women got one (3.2%). Similarly, cavity drills and fillings were performed on 108 men (3.6%) and 59 women (5.4%). Within the scope of radiography more men (*n* = 198, 6.6%) than women (*n* = 59, 5.4%) were X-rayed. More details are given in Table 3.

### 3.4. Assumption of Costs

Overall, 102 requests for the assumption of costs, according to § 6 AsylbLG were counted. Thereof, 73 were approved, 18 were rejected, and 11 outcomes were unknown (71.6%, 17.6%, and 10.8% respectively). 71.6% of the requests contained root canal therapy, of which 86.3% were accepted, and 6.9% were refused.

In contrast, 72.7% of the claims for detachable prosthesis were declined. In addition, two out of three crowns and the one requested bridgework were turned down. All applications on prosthodontic adjustment (*n* = 2), translator work (*n* = 2), and coinsurance exemption (*n* = 2) were accepted. More details are given in Figure 3.

## 4. Discussion

This study aimed at describing dental care utilization and dental treatment of asylum-seekers. Hereby, our results are in line with findings already described in the literature: As in the German multicentre cross-sectional study [10], pain is a very common complaint and the leading cause for dental consultations. Estimating the prevalence of tooth pain in the population, a prevalence of roughly 10% of all asylum-seekers was calculated. This is higher than in the previously mentioned study, where 5% of patients reported suffering pain. The higher prevalence in our cohort might be explained by different modes of data collection. In the comparative study, acute pain was defined as “existing pain with an acute need for treatment” at the time of the cross-sectional examination, while the prevalence given in our study relates to a period of one year. It should also be mentioned that there might a reporting bias because the AsylbLG emphasizes the treatment of pain over other treatment reasons, which might lead to overreporting of pain in patient files.

The most frequent underlying reasons for the patients’ complaints are caries, poor oral hygiene, and periodontal illness. This finding converges with other studies in western countries [26,27,28].

When estimating the minimal prevalence of caries in the whole population of asylum-seekers, our prevalence is much smaller compared to other studies. Solyman et al. [11], for instance, found a caries-prevalence of 80%. Thus, it was assumed that the prevalence of caries diagnosed in the health care sector greatly underestimates the prevalence of caries in the population and assumes a substantial number of unreported cases who did not report to the dentist even though they have caries.

This might be attributed to various factors: In the international literature, underdeveloped healthcare systems in countries of origin, challenges of understanding the new healthcare system, and language barriers are found to be the most important barriers to access [26,29,30,31]. They restrict the utilization at the patient, provider, and system-level [32], and complicate access by means of social isolation [33]. In addition, “a general low emphasis on oral health and promotion during the resettlement period” [11] has been noted. In addition, the common approach to resettling asylum-seekers in rural and remote areas where access to care is already poor further complicates their access to (dental) health care. Crocombe et al. showed, for instance, for the Australian context that people living in areas outside of the capital were less likely to have seen a dentist in the past 12 months [34]. Especially in the case of vulnerable populations such as asylum-seekers, these social determinants of oral health are known to conspire to restrict patients’ agency in health-related choices [35] and should, therefore, be the prime target for public health interventions [36].

Similarly, for the German context, different studies also have shown that a large number of access barriers such as language difficulties (especially related to seeking health care), limited transportation options, and isolation make it difficult for asylum-seekers to receive dental help in the German health system [13,33,37]. As already outlined in the introduction, the legal situation entails a number of administrative barriers that make access to dental care more difficult for asylum-seekers compared to regularly insured patients. An exploratory study from Germany reports that the “paper chase” made it almost impossible for patients to see a doctor immediately [37].

In sum, these factors conspire to make access to dental care very difficult for many patients and increase the risk that potential patients are not treated. According to a cross-sectional study, about “Assessment of oral health and cost of care for a group of refugees in Germany”, this creates discrimination in the health care of asylum-seekers [13].

Another issue highlighted in our study is preventive care. Check-ups at the dentist are an important precautionary measure for the prevention and early detection of possible tooth, mouth, and jaw diseases [38]. The different preventive care measures vary according to the age of the patient: Children aged between 6 and 17 years should receive individual dental prophylaxis services. They include checking of oral hygiene, inspection of the condition of the gum, thorough removal of soft dental plaque, and fissure sealing of the molars. For patients over eighteen years of age, the prophylaxis includes a half-yearly dental check-up and the removal of calculus [39].

The number of used medical services in the range of prophylaxis is lower among asylum-seekers than among patients in the statutory health insurance: 72% of all regularly insured people made a claim on at least one dentist consultation per year [40], whereas only 11% of asylum-seekers did. Just as well, the difference between prophylactic services is obvious. Compared to 66% of children in the statutory health insurance, only 4% of the children of asylum-seekers received preventive medical examinations. Similarly, among adults, only 2% of asylum-seekers—compared to 49% of regularly insured adults—received preventive check-ups.

A qualitative study from the United States explored through focus groups, how cultural factors affected access to preventive oral health care among ethnic minority groups, and what those cultural factors are. The authors see the main challenge in dental-related health literacy and concluded a “[l] ack of knowledge and beliefs about primary teeth [which] created barriers to early preventive care in all groups” [41]. The concept of routine preventive visits for teeth was not well-established among the participants. They held the view that just incipient complaints necessitate a consultation at a dentist [41]. Moreover, the majority belief assumed that primary teeth have a limited function and dispensable importance. Additionally, stress and fear resulting from previous dental experiences came restrictive to the fore [41]. A cross-sectional study from Germany [11] describes the opinion of asylum-seekers about the relationship between oral and general health. In the process, they determined that “less than one third believed in the relationship (…) and less than half believed they should have regular check-ups by a dentist”. Precautionary care also includes self-performed oral hygiene measures. The research showed that they were low among the participants [11].

For the medical care of the asylum-seekers, it is important that the suppliers have clarity about which treatments in the health service they get refunded. Because doctors often are unsure about the legal context concerning cost absorption for asylum-seekers’ health care [42], they are unsettled under which conditions costs for dental treatments will be reimbursed by the department of social security [43]. We hypothesize that as a result, acute pain treatments were given priority, and dental prevention was neglected. Considering our findings and the literature, this seems to be part of a vicious cycle: Preventive check-ups could be an important step to change asylum-seekers’ health literacy and sensitize them for the benefits of such check-ups, but if providers do not offer these check-ups for reasons of cost absorption this chance will be missed and utilization will remain low. Considering that the efficacy of dental prevention is proven in a number of studies [13,44], the attempt to save money by cutting down on prevention might turn out to be very costly for the health care system in the long run.

Our study also focuses on carious diseases and their therapy. The number of fillings applied to regularly insured patients is 28%, 7 times as high as compared to asylum-seekers. In addition, the share of 9% of extractions in regularly insured patients is about 3 times as high as the corresponding number in asylum-seekers. The comparatively low treatment frequency depicts a contrast to the high count of diagnosed carious diseases.

A report by the health insurance company BARMER assumes that a filled tooth previously was infested by caries. Thus, the authors interpret the prevalence of people with at least one filling as an indicator for the occurrence of caries requiring treatments [40].

The assumption that the number of placed fillings can be used to determine the number of caries cases cannot be confirmed in this case: If a dentist detects a carious disease, there are several ways to treat it. The treatment depends on the severity of the caries. If the tooth is still in good condition, a filling can be placed [45,46]. A root canal treatment is recommended for deep caries that has reached the pulp. In some cases, the tooth may be so badly damaged by the caries that the tooth must be extracted. The decision about the therapy is made by the attending dentist. The aim of a dentist is the preservation of teeth. Thus, not every caries can be treated with a filling. In case of deep caries with pulp opening, a root canal treatment must be administered. “Teeth not worthy of preserving” are usually drawn due to their severe carious infestation. Both treatments do not contain any filling therapy. Therefore, the BARMER report systematically underestimates the prevalence of caries. The following section presents arguments for overestimating and underestimating caries prevalence. A factor is whether detected forms of caries are treated at all. A carious disease is not always accompanied by pain [47]. Patients refuse treatment in such cases because they believe that without pain, no therapy is necessary [41]. In addition, the costs for tooth-colored filling materials are higher than metal-colored fillings [48,49]. In the anterior tooth area, the social welfare office covers the costs for tooth-colored composite fillings. In the posterior region, the costs for amalgam fillings are covered. In spite of the proven survival superiority of metal-colored fillings in some clinical scenarios [45], the current study found some patients who refuse this treatment option. The refusal is due to the controversies regarding mercury safety [50] and the aesthetic look [48,51]. It happens that patients do not attend another appointment after the diagnosis has been made. Reasons for this are, for example, a removal or even fear of treatment [52].

Before certain treatments, dentists must obtain approval from the social welfare office to cover the costs. The social welfare office´s decision on whether to approve or reject advance cost estimates for dental treatment varies: It is noticeable that the costs of some patients were covered and others were not, even though they concerned the same procedures.

The Bundeszahnärztekammer (BZÄK) published an information sheet providing an overview of the underlying legal regulations and general information on billing and treatment: It highlights, that the efforts of the responsible authorities to save costs should not impair dental health care [4]. The Kassenzahnärztliche Bundesvereinigung (KZBV) also criticizes the legal exception of asylum-seekers from regular care procedures and reprimands that case workers are not adequately trained to assess the need for medical treatment and, therefore, should not be in charge to make such potentially far-reaching decisions.

Through § 6 and the associated assumption of “all other costs”, the usage of the AsylbLG is a question of interpretation. Different from other federal states, for example, Bavaria, Saxony-Anhalt, had no official guidelines with approved dental treatments until 2017 [53]. Some therapy procedures could only be claimed after previous consultation and approval by the social welfare office. This procedure implies the danger of the development of different approval practices by individual caseworkers and between different social welfare offices [54]. Thilo Fehmel holds the same opinion in his report “The scope for decision in Social benefits law” and refers to the diversity of implementation practice in the regional interpretation of the AsylbLG [55]. The role of caseworkers’ discretion is also reflected in our data: While most claims for cost absorption were approved, some are not, even for the same complaints.

### Limitations

Our study was limited by the fact that it only contains data from the available medical reports. The data were written by different people and with different levels of detail. Some treatments were done by students and others by dentists since the university clinic serves research and teaching. There is a possibility that treatments have been carried out, which were intended for the students to practice but which would normally not have been paid for by the social welfare office.

Some age groups are underrepresented in our cohort. Only a few patients were older than 50 years of age, thus we grouped them together. Children under the age of 5 were also poorly represented. The number of women was less than the number of men. The middle age groups were strongly represented.

## 5. Conclusions

The utilization of dental care is lower among asylum-seekers than among regularly insured patients. We assume that the low prevalence rates in our data indicate existing access barriers to the German health system. As a result, not all symptomatic asylum-seekers have access to dental care. Our study depicts that pain is the most common reason for asylum-seekers to visit a dentist, while prophylaxis is virtually absent. The health care system should increase the use of preventive care, in which elements of education to improve patients’ health literacy and governmental efforts to actively lower access barriers are essential. Beyond that, our findings highlight that further efforts are needed to reduce administrative barriers in access to dental care.

## Figures and Tables

**Figure 1 ijerph-17-02672-f001:**
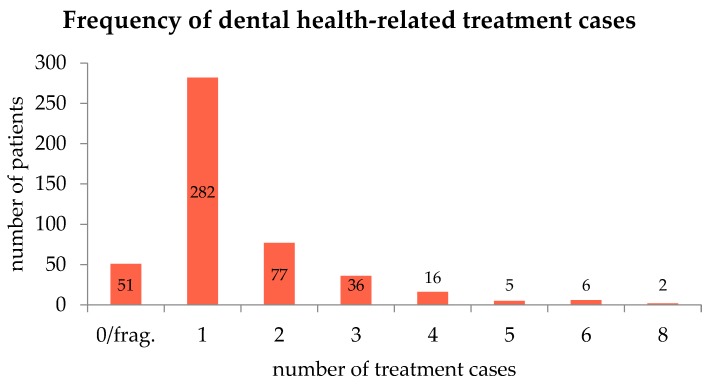
Frequency of treatment cases for dental care. Most patients had only one treatment case.

**Figure 2 ijerph-17-02672-f002:**
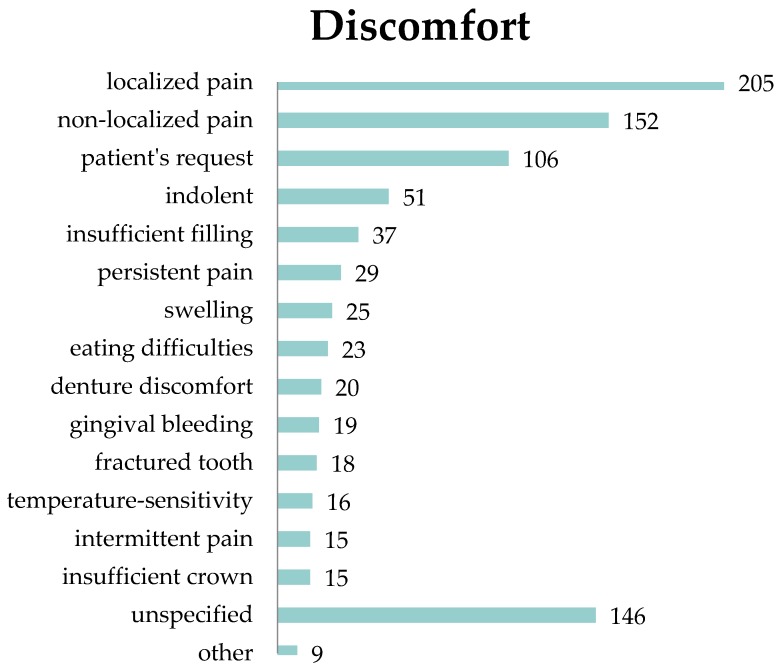
Absolute frequency of dental discomfort. Pain is the most common complaint.

**Figure 3 ijerph-17-02672-f003:**
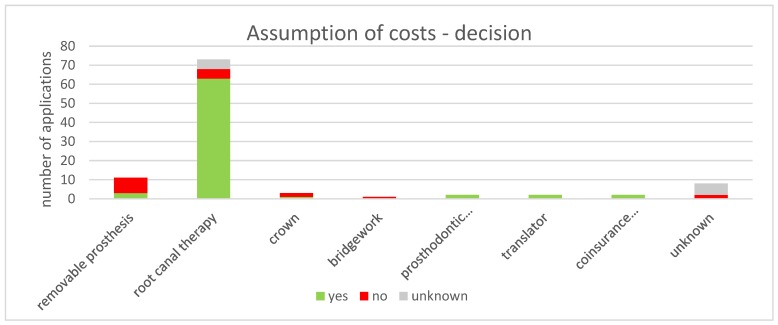
Decisions by the social welfare office to cover the cost of dental treatment applications. The majority of claims concerns root canal therapy and most of them were approved.

**Table 1 ijerph-17-02672-t001:** Demographic characteristics of the study population.

Characteristic		All Patients *n* = 475	All Asylum-Seekers *n* = 4107
Gender, *n* (%)	Female	134 (28.2)	1103 (26.9)
	Male	341 (71.8)	3004 (73.1)
Country of origin, *n* (%)	Syria	187 (39.4)	1957 (47.7)
	Afghanistan	46 (9.7)	354 (8.6)
	Iran	38 (8.0)	180 (4.4)
	Somalia	29 (6.1)	173 (4.2)
	Guinea-Bissau	21 (4.4)	116 (2.8)
	Russian Federation	21 (4.4)	98 (2.4)
	Eritrea	18 (3.8)	83 (2.0)
	India	14 (3.0)	121 (2.9)
	Kosovo	14 (3.0)	88 (2.1)
	Benin	11 (2.3)	168 (4.1)
	unknown	4 (0.8)	28 (0.7)
	others	72 (15.2)	741 (18.0)
Age group in years, *n* (%)	0 ≤ 5	6 (1.3)	322 (7.9)
	5 ≤ 10	25 (5.3)	238 (5.8)
	10 ≤ 15	14 (3.0)	171 (4.2)
	15 ≤ 20	41 (8.6)	467 (11.4)
	20 ≤ 25	86 (18.1)	795 (19.4)
	25 ≤ 30	99 (20.8)	746 (18.2)
	30 ≤ 35	71 (15.0)	535 (13.0)
	35 ≤ 40	55 (11.6)	340 (8.3)
	40 ≤ 45	28 (5.9)	195 (4.8)
	45 ≤ 50	29 (6.1)	147 (3.6)
	> 50	21 (4.4)	151 (3.7)

**Table 2 ijerph-17-02672-t002:** Frequency of asylum-seekers’ diagnoses.

Diagnosis	Subgroup	Absolute Frequency	Proportion in All Patients [%] (95%-CI)	Proportion in All Asylum–Seekers [%] (95%-CI)
Caries		469	98.7 (97.7–99.7)	11.4 (8.5–14.3)
	Primary caries	100	21.1 (17.4–24.8)	2.4 (1.0–3.8)
	Secondary caries	51	10.7 (7.9–13.5)	1.2 (0.2–2.2)
	Incipient caries	2	0.4 (0–1.0)	0.1 (0–0.4)
	Carious lesion	101	21.3 (17.6–25.0)	2.5 (1.1–3.9)
	Moderate caries	12	2.5 (1.1–3.9)	0.3 (0–0.8)
	Deep caries	107	22.5 (18.7–26.3)	2.6 (1.2–4.0)
	Deep complicated caries	96	20.2 (16.6–23.8)	2.3 (1.0–3.6)
Periodontitis ^1^		115	24.2 (20.3–28.1)	2.8 (1.3–4.3)
	Apical periodontitis	80	16.8 (13.4–20.2)	2.0 (0.7–3.3)
	Marginal periodontitis	25	5.3 (3.3–7.3)	0.6 (0–1.3)
	Periodontitis (unclassified)	10	2.1 (0.8–3.4)	0.2 (0–0.6)
Pulpitis		67	14.1 (11.0–17.2)	1.6 (0.5–2.7)
	Irreversible pulpitis	64	13.5 (10.4–16.6)	1.6 (0.5–2.7)
	Reversible pulpitis	3	0.6 (0–1.3)	0.1 (0–0.4)
Gingivitis		37	7.8 (5.4–10.2)	0.9 (0.1–1.7)
Defective restorations		108	22.7 (18.9–26.5)	2.6 (1.2–4.0)
Root canal prepared		102	21.5 (17.8–25.2)	2.5 (1.1–3.9)
Poor teeth conditions		275	57.9 (53.5–62.3)	6.7 (4.5–8.9)
	Teeth not worthy of preserving ^2^	139	29.3 (25.2–33.4)	3.4 (1.8–5.0)
	Retained/Remnant root	74	15.6 (12.3–18.9)	1.8 (0.6–3.0)
	Insufficient dentition	38	8.0 (5.6–10.4)	0.9 (0.1–1.7)
	Lack of oral hygiene	24	5.1 (3.1–7.1)	0.6 (0–1.3)
Prosthetic replacement		17	3.6 (1.9–5.3)	0.4 (0–1.0)
inflammation		32	6.7 (4.5–8.9)	0.8 (0–1.6)
	Gumboil	11	2.3 (1.0–3.6)	0.3 (0–0.8)
	Cyst	6	1.3 (0.3–2.3)	0.2 (0–0.6)
	Deep periodontal pocket	7	1.5 (0.4–2.6)	0.2 (0–0.6)
	fistula	8	1.7 (0.5–2.9)	0.2 (0–0.6)
Accretion		50	10.5 (7.7–13.3)	1.2 (0.2–2.2)
	Plaque	24	5.1 (3.1–7.1)	0.6 (0–1.3)
	Calculus	26	5.5 (3.4–7.6)	0.6 (0–1.3)
Miscellaneous		311	65.5 (61.2–69.8)	7.6 (5.2–10.0)
	Aftertreatment	138	29.1 (25.0–33.2)	3.4 (1.8–5.0)
	Uncomplaining	61	12.8 (9.8–15.8)	1.5 (0.4–2.6)
	Unspecified	26	5.5 (3.4–7.6)	0.6 (0–1.3)
	No communication possible	25	5.2 (3.2–7.2)	0.6 (0–1.3)
	Other	61	12.8 (9.8–15.8)	1.5 (0.4–2.6)

^1^ There was no consistent subdivision into “chronic or acute”/“localized or generalized” forms in the medical records. Therefore, we only differentiated according to the cause of origin. ^2^ There are several reasons why a tooth is classified as “not worth preserving,” such as carious destruction, root infection of a dead tooth, or a tooth fracture. A clear assignment is not possible; therefore, an additional category was created.

**Table 3 ijerph-17-02672-t003:** Frequency of treatment procedures *.

	Absolute			All Patients (%)	All Asylum-Seekers (%)		
	Male	Female	Total	Total	Male	Female	Total
Clinical examination and consultation	310	125	435	91.6	10.3	11.3	10.6
Radiography	198	59	257	54.1	6.6	5.4	6.3
Prophylaxis (all)	55	35	90	19.0	1.8	3.2	2.2
Prophylaxis (age 6–17 years)	14	9	23	51.1	4.3	4.5	4.4
Tooth preparation and filling	108	59	167	35.2	3.6	5.4	4.1
Pulp and root canal treatment	122	47	169	35.6	4.1	4.3	4.1
Extraction	100	34	134	28.2	3.3	3.1	3.3
Surgical intervention	84	27	111	23.4	2.8	2.5	2.7
Minimal intervention	45	22	67	14.1	1.5	2.0	1.6
Anaesthesia	203	71	274	57.7	6.8	6.4	6.7

* The uniform assessment standard for dental services (BEMA) is divided into five parts. Part one (preservative and surgical performance and x-ray performance) is described in detail here. No information was provided in the available files on services from part two (broken jaw, TMJ disorders) and part four (systematic treatment of periodontal disease). Only one patient was treated with services from part three (orthodontic treatment). A small proportion of the patients received services according to part five (restoration of dentures and dental crowns).

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
