# Peer review of "Dental Care for Asylum-Seekers in Germany: A Retrospective Hospital-Based Study"

_ijerph, 2020, doi:10.3390/ijerph17082672_

Round 1

Reviewer 1 Report

The introduction is well written, method section is well developed. However, i find gaps in discussion section.

Discussion

In the paragraph "This might be attributed to various factors: In the international literature underdeveloped...... "It would be essential to develop argument on addressing social determinants of health "cause of the cause" reducing the gap by Marmot et al.,

Also asylum seekers are mostly placed in rural and remote areas where access to care is already poor. You can reference Crocombe et al., paper on rural versus urban differences in oral health care in Australia and also comparative paper between germany and australia.

Also stress, depression, anxiety are contributors to dental problems. Could highlight that too.

Conclusion:

In conclusion of the study. the author talks about access to care as a limitation for having high dental caries and periodontitis in Asylum seekers. Also the dental visits were associated with pain. In my understanding elements of education, behaviour change, government support are essential components that need to addressed. Especially in situations like COVID-19 a pandemic plan is required for dental care seekers [because of low social determinants of health]. 

Author Response

Response to Reviewer 1 Comments

Response: Dear Sir or Madam, we would like to thank you for the helpful comments. Below you find our comments concerning the details of the revision.

The introduction is well written, method section is well developed. However, I find gaps in discussion section.

Discussion

Point 1: In the paragraph "This might be attributed to various factors: In the international literature underdeveloped...... "It would be essential to develop argument on addressing social determinants of health "cause of the cause" reducing the gap by Marmot et al.,

Response 1: We thank the reviewer for this hint and added a more explicit discussion of the social determinants of health to the discussion section.

Point 2: Also asylum seekers are mostly placed in rural and remote areas where access to care is already poor. You can reference Crocombe et al., paper on rural versus urban differences in oral health care in Australia and also comparative paper between germany and australia.

Also stress, depression, anxiety are contributors to dental problems. Could highlight that too.

Response 2: We agree with the reviewer and revised the text accordingly.

Conclusion

Point 3: In conclusion of the study, the author talks about access to care as a limitation for having high dental caries and periodontitis in Asylum seekers. Also the dental visits were associated with pain. In my understanding elements of education, behaviour change, government support are essential components that need to addressed. Especially in situations like COVID-19 a pandemic plan is required for dental care seekers [because of low social determinants of health].

Response 3: We certainly agree with the reviewer and included a brief discussion of issues of health literacy in the conclusions section.

Reviewer 2 Report

Interesting paper. Some concerns:

  • To facilitate reading, this referee recommends placing an ethical committee at the beginning of the Material and methods section.
  • How was the sample size chosen? Has a power analysis been carried out?
  • Authors should be inclusion and exclusion criteria
  • Authors excessively use "we .... recommend using the impersonal form
  • Why only descriptive analysis? I recommend other statistical analysis
  • The whole discussion is poorly written and insufficient. The discussion is also misleading
  • This referee does not understand “[…]. appearing in some parts of the text

Author Response

Response to Reviewer 2 Comments

Response: Dear Sir or Madam, we would like to thank you for the helpful comments. Below you find our outline concerning the details of the revision.

Interesting paper. Some concerns:

Point 1: To facilitate reading, this referee recommends placing an ethical committee at the beginning of the Material and methods section.

Response 1: As suggested by the reviewer, we moved information on the ethics approval to the beginning of the methods section.

Point 2: How was the sample size chosen? Authors should be inclusion and exclusion criteria

Response 2: We added a paragraph to the methods section that outlines some more details concerning inclusion and exclusion criteria. Since this study included all eligible participants, no sample size calculation was done in advance.

Point 3: Authors excessively use "we .... recommend using the impersonal form

Response 3: In our choice for active voice we follow the stylistic guidelines by APA and other sources (e.g. Hofmann 2010: Scientific Writing and Communication. Oxford University Press).Nevertheless, to avoid the impression of an excessive use of active voice we changed the phrasing into impersonal forms wherever it was stylistically possible.

Point 4: Has a power analysis been carried out? Why only descriptive analysis? I recommend other statistical analysis

Response 4: Recently published recommendations of the American Statistical Association (Wasserstein et al. (2019), The American Statistician 73, 1-19; Amrhein et al. (2019), Nature 567, 305-307) discourage the use of statistical tests in exploratory approaches. Since our study design is of exploratory nature we did not employ any statistical test and therefore did not perform any power analysis. We added a sentence to the methods section where we explain this in more detail.  

Point 5: The whole discussion is poorly written and insufficient. The discussion is also misleading

Response 5: We thank the reviewer for his or her comments concerning the discussion section and took the opportunity to rework this part of the manuscript.

Point 6: This referee does not understand “[…]. appearing in some parts of the text

Response 6: We deleted the needless “[…]” at the beginning of the quotations for better readability.

Reviewer 3 Report

The submitted manuscript was read with interest.
There are a few recommendations that must be taken into consideration to enhance the manuscript.

Discussion section.
- After the sentence "If the tooth is still in good condition, a filling can be placed," add the references J Prosthet Dent. 2017 Mar;117(3):345-353.e8, as well as Int Endod J. 2017 Oct;50(10):951-966.
- Replace the sentence "Some patients refuse the use of metal-coloured fillings." with the following sentence "In spite of the proven survival superiority of metal-coloured fillings in some clinical scenarios, the current study found some patients who refuse this treatment option." However, insert the reference "J Prosthet Dent. 2017 Mar;117(3):345-353.e8" after "clinical scenarios."
- Page 8. Please avoid using a footnote comment. This information may add it in another section of the text or move it as supplemental material.

Author Response

Response to Reviewer 3 Comments

Response: Dear Sir or Madam, we would like to thank you for the helpful comments. Below you find our comments concerning the details of the revision.

The submitted manuscript was read with interest.

There are a few recommendations that must be taken into consideration to enhance the manuscript.

Discussion section

Point 1: After the sentence "If the tooth is still in good condition, a filling can be placed," add the references J Prosthet Dent. 2017 Mar;117(3):345-353.e8, as well as Int Endod J. 2017 Oct;50(10):951-966.

Response 1: We thank the reviewer for drawing our attention to these articles. We added the references to the manuscript.

Point 2: Replace the sentence "Some patients refuse the use of metal-coloured fillings." with the following sentence "In spite of the proven survival superiority of metal-coloured fillings in some clinical scenarios, the current study found some patients who refuse this treatment option." However, insert the reference "J Prosthet Dent. 2017 Mar;117(3):345-353.e8" after "clinical scenarios."

Response 2: We thank the reviewer for this specification of the argument and revised the text accordingly.

Point 3: Page 8. Please avoid using a footnote comment. This information may add it in another section of the text or move it as supplemental material.

Response 3: We thank the reviewer for this hint. The footnote was indeed wrongly placed and is now moved to its correct location as a legend to Table 3. The same applies for footnotes 1 and 2 which are legends to Table 2.